# Prevalence and antimicrobial resistance pattern of *Clostridium difficile* among hospitalized diarrheal patients: A systematic review and meta-analysis

**Tebelay Dilnessa**[1,2]*, **Alem Getaneh**[1], **Workagegnehu Hailu**[3], **Feleke Moges**[1], **Baye Gelaw**[1]

1 Department of Medical Microbiology, School of Biomedical and Laboratory Sciences, College of Medicine and Health Sciences, University of Gondar, Gondar, Ethiopia, 2 Department of Medical Laboratory Sciences, College of Health Sciences, Debre Markos University, Debre Markos, Ethiopia, 3 Department of Internal Medicine, School of Medicine, College of Medicine and Health Sciences, University of Gondar, Gondar, Ethiopia

* tebelay@gmail.com

**Data Availability Statement:** All relevant data are within the manuscript and its Supporting information files.

## Abstract

### Background

*Clostridium difficile* is the leading cause of infectious diarrhea that develops in patients after hospitalization during antibiotic administration. It has also become a big issue in community-acquired diarrhea. The emergence of hypervirulent strains of *C. difficile* poses a major problem in hospital-associated diarrhea outbreaks and it is difficult to treat. The antimicrobial resistance in *C. difficile* has worsened due to the inappropriate use of broad-spectrum antibiotics including cephalosporins, clindamycin, tetracycline, and fluoroquinolones together with the emergence of hypervirulent strains.

### Objective

To estimate the pooled prevalence and antimicrobial resistance pattern of *C. difficile* derived from hospitalized diarrheal patients, a systematic review and meta-analysis was performed.

### Methods

Preferred Reporting Items for Systematic Reviews and Meta-Analyses (PRISMA) guideline was followed to review published studies conducted. We searched bibliographic databases from PubMed, Scopus, Google Scholar, and Cochrane Library for studies on the prevalence and antimicrobial susceptibility testing on *C. difficile*. The weighted pooled prevalence and resistance for each antimicrobial agent was calculated using a random-effects model. A funnel plot and Egger's regression test were used to see publication bias.

### Results

A total of 15 studies were included. Ten articles for prevalence study and 5 additional studies for antimicrobial susceptibility testing of *C. difficile* were included. A total of 1967/7852

**Funding:** The author(s) received no specific funding for this work.

**Competing interests:** The authors have declared that no competing interests exist.

(25%) *C. difficile* were isolated from 10 included studies for prevalence study. The overall weighted pooled proportion (WPP) of *C. difficile* was 30% (95% CI: 10.0–49.0; p<0.001). The analysis showed substantial heterogeneity among studies (Cochran's test = 7038.73, $I^2$ = 99.87%; p<0.001). The weighed pooled antimicrobial resistance (WPR) were: vancomycin 3%(95% CI: 1.0–4.0, p<0.001); metronidazole 5%(95% CI: 3.0–7.0, p<0.001); clindamycin 61%(95% CI: 52.0–69.0, p<0.001); moxifloxacin 42%(95% CI: 29–54, p<0.001); tetracycline 35%(95% CI: 22–49, p<0.001); erythromycin 61%(95% CI: 48–75, p<0.001) and ciprofloxacin 64%(95% CI: 48–80; p< 0.001) using the random effect model.

## Conclusions

A higher weighted pooled prevalence of *C. difficile* was observed. It needs a great deal of attention to decrease the prevailing prevalence. The resistance of *C. difficile* to metronidazole and vancomycin was low compared to other drugs used to treat *C. difficile* infection. Periodic antimicrobial resistance monitoring is vital for appropriate therapy of *C. difficile* infection.

## Introduction

*Clostridium difficile* infection (CDI) is a major problem as a healthcare-associated infection that occurs mainly in conjunction with the use of broad-spectrum antibiotics [1]. It is responsible for 15–25% of cases of antibiotic-associated diarrhea and virtually all cases of antibiotic-associated pseudomembranous colitis [2]. CDI has currently exceeded methicillin-resistant *Staphylococcus aureus* (MRSA) as hospital-acquired infections [3] and mortality associated with *C. difficile* infectious diarrhea is estimated to be 17% and up to 25% among the elderly [4]. A significant increase in the incidence of *C. difficile* associated diarrhea gained the great interest for public health in the United States, Canada, and European countries due to the consequence of its pathogenicity and virulence, especially in the nosocomial field [5]. In 2008, an estimated 1 million cases of CDI have occurred [6] and are also responsible for an estimated 250,000 illnesses and 14,000 deaths per year in the United States [7].

This organism is a Gram-positive, anaerobic spore-forming bacillus that is usually spread by the fecal-oral route. It is non-invasive and produces two important exotoxins, toxin A and B that cause disease, ranging from asymptomatic carriage to mild diarrhea, to colitis, or pseudomembranous colitis, fulminant colitis, and toxic megacolon [8, 9]. *C. difficile* is ubiquitous and widely distributed in nature. It produces infectious spores that are highly resistant to disinfection and harsh environments, potentially facilitating spread over distance and nosocomial transmission of *C. difficile* [10]. Patients can be contaminated from the environment surface, share instrumentation, hospital personnel hand, and infected roommate. In addition to exogenous, the source of infection also may from an endogenous source due to the presence of the organism as normal flora in the intestine. It is considered part of the normal flora of infants and can be isolated from 3–5% of healthy adults and 16–35% in asymptomatic or colonized hospitalized patients [11].

Outbreaks of CDI are linked to the emergence of hypervirulent drug-resistant strains. Concurrently, misuse or wrong use of treatment alternatives can result in clinical difficulties comprising the occurrence of antibiotic resistance and growing rates of diseases associated with antibiotic usage [12]. The undifferentiating habits of the use of broad-spectrum antibiotics

upset the balance of the normal intestinal microbiota and weakens colonization resistance [13], thereby providing a niche for colonization by *C. difficile* producing toxins A and B [14]. They function as glucosyltransferases that inactivate Rho, Rac1, and Cdc42 within host cells, leading to actin polymerization, the opening of tight junctions, and ultimately cell death [15]. The rate of acquisition of CDI increases linearly with the length of hospital stay and can reach 40% after 4 weeks of hospitalization [16–18]. The main antibiotics that are associated with CDI are clindamycin and extended-cephalosporins and fluoroquinolone [19].

The occurrence and spread of *C. difficile* isolates resistant to several antibiotics, particularly among the hypervirulent *C. difficile* ribotype 027 strains, are now a growing problem for the management of *C. difficile* infections [20]. The antimicrobial resistance in *C. difficile* has worsened due to the inappropriate use of broad-spectrum antibiotics including cephalosporins, clindamycin, tetracycline, and fluoroquinolones as well as bacterial adaptations that drive the evolution for resistance [21]. It is mostly accompanied by the acquisition of mobile genetic elements. *C. difficile* could inactivate drugs that enter a cell either by degrading or modifying them into a non-functional form by both enzymatic degradation and modification [22].

At this time, three antimicrobial agents including, metronidazole, vancomycin, and fidaxomicin are commended for the management of *C. difficile* infection and numerous novel anti-CDI antibiotics are in clinical trials [23]. Additionally, many antibiotics such as fidaxomicin, fusidic acid (FDA), and rifamycins (RIFs) were introduced to fight *C. difficile* infections, but the resistance to these drugs after treatment has also been recognized [24].

It is important to obtain information about the burden of CDI and resistance profiles of circulating *C. difficile* strains. This review aimed to determine the following inquiries: i) what is the pooled prevalence of *C. difficile* among studies included in those hospitalized diarrheal patients worldwide? ii) what is the weighted pooled resistance of *C. difficile* to antimicrobials used to treat it? Therefore, this review provided the pooled prevalence and antimicrobial resistance pattern of each antimicrobial against *C. difficile* infection.

## Methods

### Searching strategy and information sources

The results of this review were registered on Prospero with ID: CRD42021255134 and reported based on the Preferred Reporting Items for Systematic Review and Meta-Analysis statement (PRISMA) guideline [25]. Articles that were potentially relevant to meta-analysis and systematic review, comprehensive searches were performed in the following databases: PubMed, Scopus, Google Scholar, and Cochrane Library. All searches were limited to articles written in English given that such language restriction does not alter the outcome of the systematic reviews and meta-analyses [26].

The search strings or terms were stemmed from the following keywords: prevalence, *Clostridium difficile*, *Clostridioides*, antibiotics, antimicrobial susceptibility, and drug resistance. The search terms were used to retrieve relevant literature in a combined form adapted to the requirement of the specific database. In the advanced searching databases, the searching strategy was built based on the above-mentioned terms using the" Medical Subject Headings (MeSH)" and "All fields" by linking "AND" and "OR" Boolean operator terms as appropriate.

Search engine:- "Clostridioides difficile"[MeSH Terms] OR ("clostridioides"[All Fields] AND "difficile"[All Fields]) OR "Clostridioides difficile"[Title] OR ("clostridium"[Title] AND "difficile" [Title]) OR "clostridium difficile"[Title/Abstract] AND "antimicrobial"[All Fields] OR "antimicrobial susceptibility" [Title/Abstract] AND "drug resistance"[MeSH Terms] AND (("2007/01/01"[PDat]: "2020/12/30"[PDat])).

## Study selection and eligibility criteria

The following criteria were used to include and exclude studies from the review.

## Inclusion criteria

- Participants: this review included studies that were conducted among hospitalized diarrheal patients

- Setting: Studies conducted at the health institution level.

- Outcome: If the studies provide the prevalence of *C. difficile* and their susceptibility to antimicrobials through culture-based tests

- Type of study: only cross-sectional studies were included.

- Publication types: journal articles, master's thesis, and dissertations

- Studies published only in the English language were included in the review.

- Time frame: all studies published between January 01, 2007, and December 30, 2020, were included.

- Any study that was using the Centers for Disease Control and Prevention (CDC) criteria for prevalence and susceptibility testing of *C. difficile* were also eligible.

## Exclusion criteria

- Studies with the methodological problem and review articles were also excluded from the review.

- Studies in which the diagnosis of *C. difficile* was not based on the CDC/National Healthcare Safety Network criteria.

- Retrospective studies, case reports, case series, letters, commentaries, notes, editorials, and conference abstracts

- Studies were conducted among a very select group of patients (e.g, HIV patients) as they would not be generalizable to the entire population and were more susceptible to infection.

- Review articles, meta-analyses, or non-English studies were excluded.

## Study screening and selection processes

All articles were accessed from databases and electronics search engines were exported to endnote version 7 (Thomson Reuters, London) reference manager to remove duplicate studies. The remaining articles were also evaluated in the context of the topic and language. Then those articles that did not full fill the inclusion criteria of the review were rejected. Finally, the abstracts and the full-texts of the remaining studies were reviewed.

   **Quality assessment and risk appraisal.** Newcastle-Ottawa Quality Assessment Scale (NOS) was used to assess the study quality. The quality of each article was assessed using the Joanna Briggs Institute (JBI) critical appraisal tool prepared for cross-sectional studies [27]. Briefly, items that will be used to appraise cross-sectional studies: (1) inclusion criteria, (2)

description of study subject and setting, (3) valid and reliable measurement of exposure, (4) objective and standard criteria used, (5) outcome measurement, and (6) appropriate statistical analysis. A score ranging from 0 to 8 points was attributed to each study ($\geq$ 5 points: high quality, 3–4 points: moderate quality, $\leq$ 2 points: low quality). A higher score indicated a higher study quality. Different opinions on scoring were resolved through discussion among the research group until consensus was reached. The quality scale of primary studies will be considered as low risk for both systematic review and meta-analysis if the studies had got 50% and above. Of the studies on isolates of origins, 100% were human in the target population (S1 Checklist).

**Data extraction process.** All selected studies were extracted using a standardized data collection form. Two independent reviewers (TD and AG) were extracted the data including the name of the first author, year of publication, study area, study design, target population, sample size, *C. difficile* prevalence, isolation techniques, and antimicrobial susceptibility testing. If the prevalence was not reported directly, it was calculated using the sample size and number of outcomes.

**Outcome measures.** The primary outcome measures were the prevalence of *C. difficile* through culture and resistance pattern of *C. difficile* among standard antimicrobials. Resistance was defined according to either the European Committee on Antimicrobial Susceptibility Testing (EUCAST) [28] or the Clinical & Laboratory Standards Institute (CLSI) [29] minimal inhibitory concentration (MIC) interpretative breakpoints.

## Data processing and analysis

Data were extracted in Microsoft Excel format, followed by analysis using STATA Version 16 statistical software. Publication bias and heterogeneity across studies were assessed using the Cochrane Q test and $I^2$ statistics [30]. $I^2$ heterogeneity test, in which 0–40%, 50–60%, 50–90%, and 75–100% represented low, moderate, substantial, and considerable heterogeneity, respectively [30, 31]. $I^2$ heterogeneity test >50% and p-value <0.01 were indicated the presence of heterogeneity and the Dersimonian laired random-effects model was employed [32]. For identification of influential studies that resulted in variation, first, the extracted data was checked for any error that might happen during the process of extraction then sensitivity analysis was carried out using the "metaninf" command [33]. Finally, if the data was free of errors and when there is no outlier using sensitivity analysis further subgroup analysis was conducted. The subgroup analyses were employed by assuming the study area and year of publication as sources of variation.

Egger's regression objectivity test was used to assess the publication bias [34]. Accordingly, the asymmetry of the funnel plot and/or statistical significance of Egger's regression test (p<0.05) was suggestive of publication bias. Therefore, the "metatrim" command, a nonparametric method of analysis was done [35]. Furthermore, all statistical interpretations were reported based on the 95% CI.

## Quality of studies and bias

The Joanna Briggs Institute (JBI) critical appraisal tool for cross-sectional studies was used to assess the methodological quality of each study. This tool was used to detect the occurrence of any real evidence of bias based on (i) target population, (ii) sampling population, and (iii) sample size [36]. The Begg and Mazmudar rank correlation test was used to assess bias across studies [37]. Since all the studies that fulfill the eligibility criteria of this systematic review and meta-analysis had got 50% and above. Therefore, all of them were considered for analysis.

## Results

### Selection and characteristics of included studies

A total of 5422 articles were searched through electronic searches of which 2814 duplicated articles were excluded. From the remaining 2608 articles, 1544 articles were excluded after reading titles and abstracts, and inaccessibility of full text. Finally, 192 full-text articles were accessed for eligibility criteria. Based on the predefined criteria and after critical appraisal 15 articles were included in the final systematic review meta-analysis (Fig 1).

All the included studies were conducted and published between January 01, 2007, and December 30, 2020. The articles which fulfill eligibility criteria for this systematic review and meta-analysis were conducted. Seven studies from Iran [12, 38–43]; two studies in China [44, 45]; one study in the United States [46]; one study in European Union (EU) [47]; one study in Poland [48]; one study in Scotland [49]; one study in South Korea [50] and lastly one study in Thailand [51]. All of these studies were done by cross-sectional study design and conducted among hospitalized diarrheal patients (Table 1).

The studies included in the meta-analysis assessed for prevalence and antibiotic resistance to vancomycin, metronidazole, moxifloxacin, clindamycin, tetracycline, erythromycin, and ciprofloxacin of *C. difficile*. A total of 1967/7852 (25%) *C. difficile* were isolated from 10 included studies for prevalence study. But for antimicrobial susceptibility testing, among 15 articles included, 2703 isolates were tested to various antimicrobials.

### Meta-analyses

### The pooled prevalence of *C. difficile*

A total of 1967/7852 (25%) *C. difficile* were isolated from 10 included studies for the prevalence study. The overall weighted pooled proportion (WPP) of *C. difficile* detection using the random-effect model was 30.0%(95%CI: 10.0–49.0; p<0.001). As shown in the forest plot graph, substantial heterogeneity was identified (Q = 7038.73, $I^2$ = 99.87; p<0.001) indicating that the use of random-effects models for estimating the pooled estimates is applicable (Fig 2). Moreover, it also suggests the need to conduct subgroup analysis to identify the sources of heterogeneity.

**Heterogeneity and publication bias.** The analysis showed substantial heterogeneity among studies on the human subject (Cochran's Q test = 7038.73, p<0.001; $I^2$ = 99.87%, p<0.001) (Fig 2). In the observational test for publication bias, a funnel plot showed an asymmetrical distribution of studies (Fig 3). Likewise, Egger's regression test with an estimated bias coefficient is -19.5 with a standard error of 4.5, giving a p-value of 0.003. The test thus provides strong evidence for publication bias (Table 2).

**Subgroup analysis.** Subgroup analysis was done based on the study area and year of publication to identify the possible source of heterogeneity across studies. The results of the subgroup analysis done by considering both country and year of publication showed still heterogeneity was considerable ($I^2$ = 99.4%, p<0.001). According to the subgroup analysis based on the country of study, the pooled prevalence of *C. difficile* in Iran and China were found to be 23.42% (95% CI: 12.23–34.61) and 19.39% (95% CI: 5.99–32.78), respectively (Fig 4). Similarly, the pooled prevalence of *C. difficile* based on the year of publication increased from earliest to latest. For example, the pooled prevalence in the year 2009, 6.05% (95%CI: 4.53–7.57; $I^2$ = 0, p>0), in 2016, 13.61% (95%CI: -1.22–28.45; $I^2$ = 96, p<0.001) and in 2019, 39.52% (95%CI: 13.11–65.93; $I^2$ = 96.1, p<0.001) (Fig 5).

**Sensitivity analysis.** Sensitivity analysis shows the robustness of the observed outcomes to the assumptions considered in executing the analysis. It indicates the influence of one study on

## PRISMA 2009 Flow Diagram

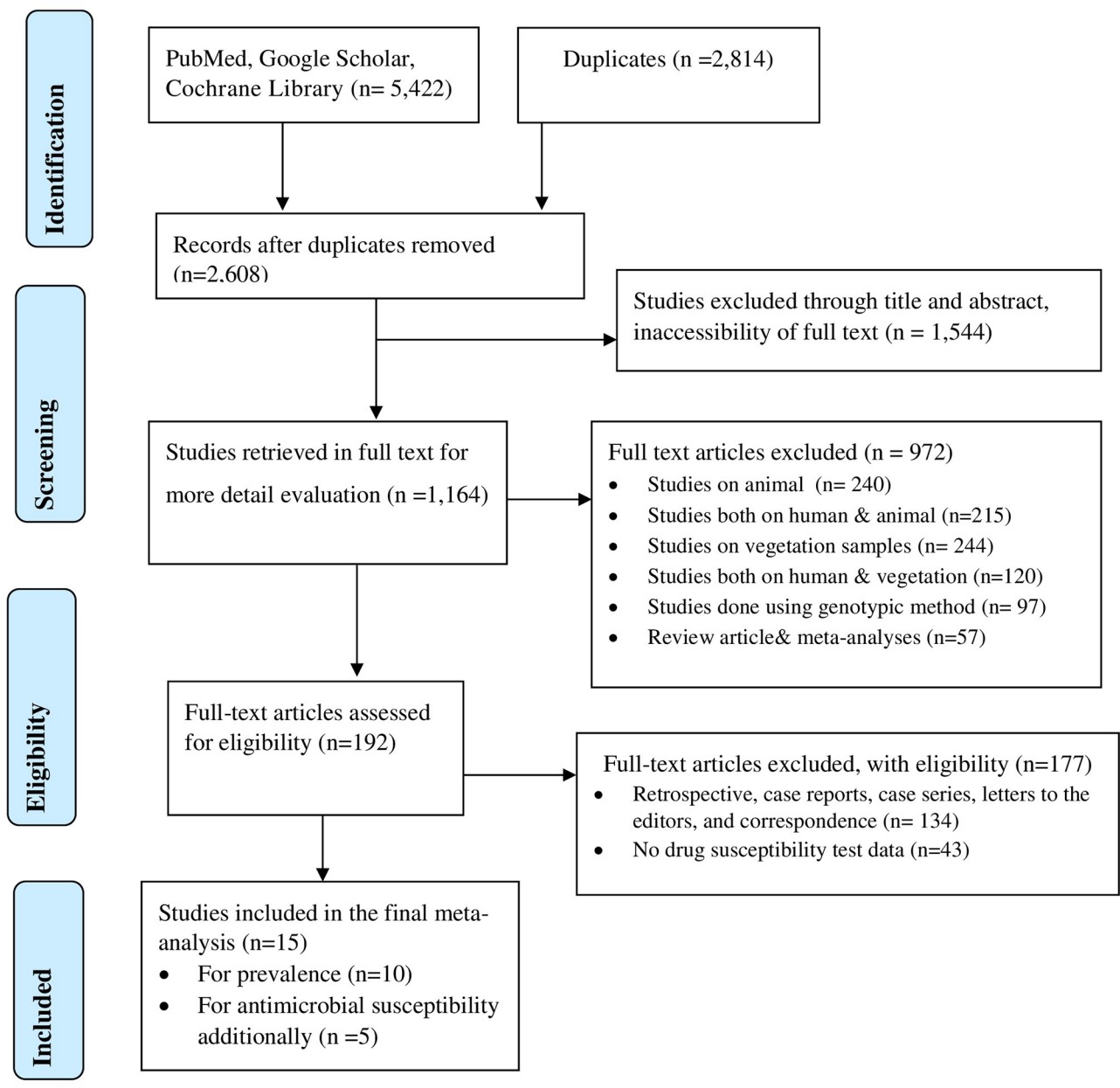

**Fig 1. Flow chart of literature search and inclusion/exclusion criteria.**

the overall meta-analysis estimates. In this review, the prevalence of *C. difficile* isolates from each study was within the confidence interval limit (S1 Fig).

## Pooled antimicrobial resistance testing of *C. difficile*

**Vancomycin resistance of *C. difficile*.** A resistance pattern of *C. difficile* to vancomycin was determined in 15 studies among 2755 isolates. Of which 114/2755 (4.1%) isolates were

**Table 1. The characteristics of included studies for prevalence and antimicrobial susceptibility of *C. difficile* among hospitalized diarrheal patients.**

| First author | Year of publication | Country | Study design | Sample size (N) | Patients with CD (n) | CD tested for susceptibility | Prevalence of CD (n/N) x100 | Reference |
|---|---|---|---|---|---|---|---|---|
| Alimolaei M | 2019 | Iran | Cross-sectional | 151 | 133 | 133 | 88.10 | [38] |
| Baghani A | 2018 | Iran | Cross-sectional | 735 | 46 | 46 | 6.26 | [39] |
| Freeman J | 2014 | EU | Cross-sectional | 953 | 866 | 866 | 90.88 | [47] |
| Goudarzi M | 2013 | Iran | Cross-sectional | 390 | 75 | 75 | 19.23 | [12] |
| Kouzegaran S | 2016 | Iran | Cross-sectional | 400 | 25 | 25 | 6.25 | [40] |
| Sedigh E-Saraie H | 2016 | Iran | Cross-sectional | 215 | 46 | 46 | 21.40 | [41] |
| Mohammadbeigi M | 2019 | Iran | Cross-sectional | 2947 | 538 | 538 | 18.25 | [42] |
| Zhou Y | 2019 | China | Cross-sectional | 839 | 107 | 73 | 12.75 | [44] |
| Wang R | 2018 | China | Cross-sectional | 280 | 74 | 74 | 26.42 | [45] |
| Sadeghifard N | 2009 | Iran | Cross-sectional | 942 | 57 | 57 | 6.05 | [43] |
| Mutlu E | 2007 | Scotland | Cross-sectional | - | 116 | 116 | - | [49] |
| Nicholas A | 2017 | S. Korea | Cross-sectional | - | 70 | 70 | - | [50] |
| Peng Z | 2017 | USA | Cross-sectional | - | 139 | 139 | - | [46] |
| Pituch H | 2011 | Poland | Cross-sectional | - | 330 | 330 | - | [48] |
| Putsathit P | 2017 | Thailand | Cross-sectional | - | 105 | 105 | - | [51] |

**Notes**: CD: *Clostridium difficile*; EU: European Union, USA; United States of America

resistant to vancomycin. The overall weighted pooled resistance (WPR) of vancomycin worldwide was 3.0% (95% CI: 1.0–4.0; p<0.001) using the random effect model (Table 3). Substantial heterogeneity was identified ($I^2$ = 89.55; p<0.001) indicating that the use of random-effects models for estimating the pooled is applicable (Fig 6). It also suggests the need to conduct subgroup analysis to identify the sources of heterogeneity.

According to the subgroup analysis based on country of study the pooled vancomycin resistance of *C. difficile* in Iran and China were found to be 1.99% (95%CI: -7.65–11.65; $I^2$ = 0%, p = 0.98) and 1.36% (95%CI: -14.59–17.42; $I^2$ = 0%, p = 0.99), respectively (S2 Fig). Similarly, the pooled vancomycin resistance of *C. difficile* based on the year of publication was observed. For example, the pooled vancomycin resistance in the year 2009, 1.75% (95%CI: -23.9–27.5; $I^2$ = 0%, p = 1), in 2016, 4.27% (95%CI: -22.6–31.12; $I^2$ = 0%, p = 0.57) and in 2019, 1.1% (95%CI: -9.43–11.58; $I^2$ = 0%, p = 0.70) (S3 Fig). But the country of the study was not the source of heterogeneity. In other words, no statistical difference was found between the data from 2007 onwards 2020 (p >0.01).

**Metronidazole resistance of *C. difficile*.** The susceptibility to metronidazole was determined in 15 studies and among 2753 *C. difficile* isolates. Of which 137/2753(4.9%) isolates were resistant to metronidazole. The overall weighted pooled metronidazole resistance using the random effect model was 5.0% (95%CI: 3.0–7.0, p<0.001). Substantial heterogeneity was identified ($I^2$ = 89.55%; p<0.001) indicating that the use of random-effects models for estimating the pooled is appropriate (Fig 7, Table 3).

**Clindamycin resistance of *C. difficile*.** A resistance pattern of *C. difficile* to clindamycin was determined in 13 studies among 2638 isolates. Of which 1550/2638 (58.7%) isolates were resistant to clindamycin. The overall weighted pooled resistance of clindamycin to *C. difficile* using the random effect model was 61.0% (95%CI: 52.0–69.0; p<0.001). Substantial heterogeneity was identified ($I^2$ = 95.1%; p<0.001) using random-effects models to estimate the pooled resistance (Table 3). According to the subgroup analysis based on country of study, the

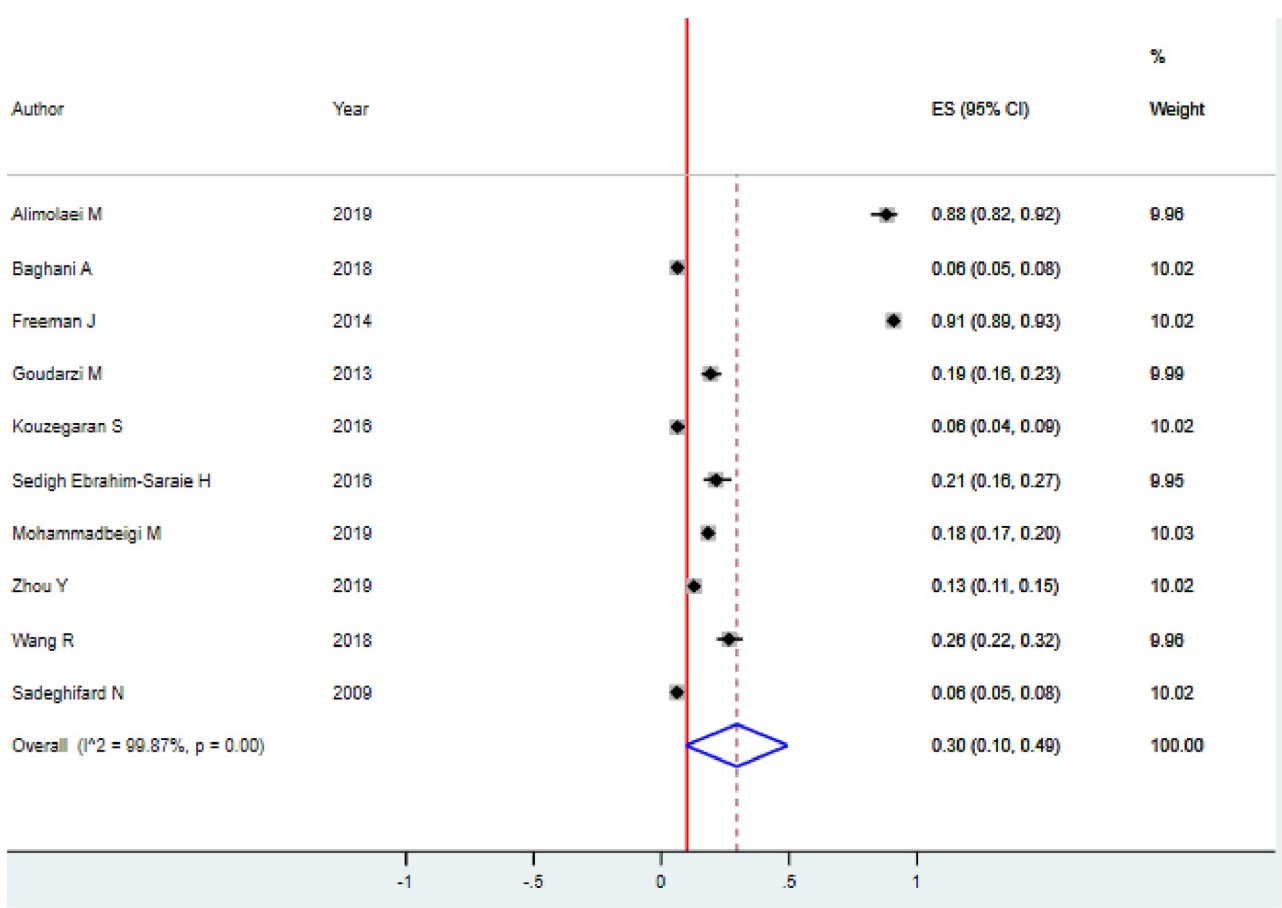

**Fig 2. Forest plot for the proportion of *C. difficile* among hospitalized diarrheal patients.**

weighted pooled clindamycin resistance of *C. difficile* in Iran and China were found to be 63.06% (95%CI: 49.55–76.57; p<0.001) and 44.87% (95%CI: 28.28–61.47; p<0.001), respectively (S4 Fig).

**Moxifloxacin resistance of *C. difficile*.**   A susceptibility to moxifloxacin was determined in 11 studies and from these studies, 2503 isolates were examined for resistance. Of these 924/2503 (36.9%) were resistant strains. The WPR to moxifloxacin was 42% (95%CI: 29–54; p<0.001) with a substantial heterogeneity ($I^2 = 97.8\%$; p<0.001) (Table 3).

**Tetracycline resistance of *C. difficile*.**   The resistance to tetracycline was determined in 10 studies among 2321 *C. difficile* isolates and from those 878/2321 (37.8%) isolates were found to be resistant. The WPR of tetracycline was 35% (95%CI: 22–49; p<0.001), with substantial heterogeneity ($I^2 = 98.3\%$; p<0.001) (Table 3). There was no significant difference between the country of data collection and year of publication ($I^2 < 16\%$ & p>0.05), but a statistically significant difference was found in Iran with weighted pooled resistance of 52% (95% CI: 33–71; p<0.001).

**Erythromycin resistance of *C. difficile*.**   The resistance to erythromycin was determined in 9 studies investigating 1544 *C. difficile* isolates. From these, 987/1544(63.9%) isolates were found to be resistant. The WPR to erythromycin was 61% (95% CI: 48–75; p<0.001) with a substantial heterogeneity ($I^2 = 97.3\%$; p< 0.001) (Table 3). The study area was not the source

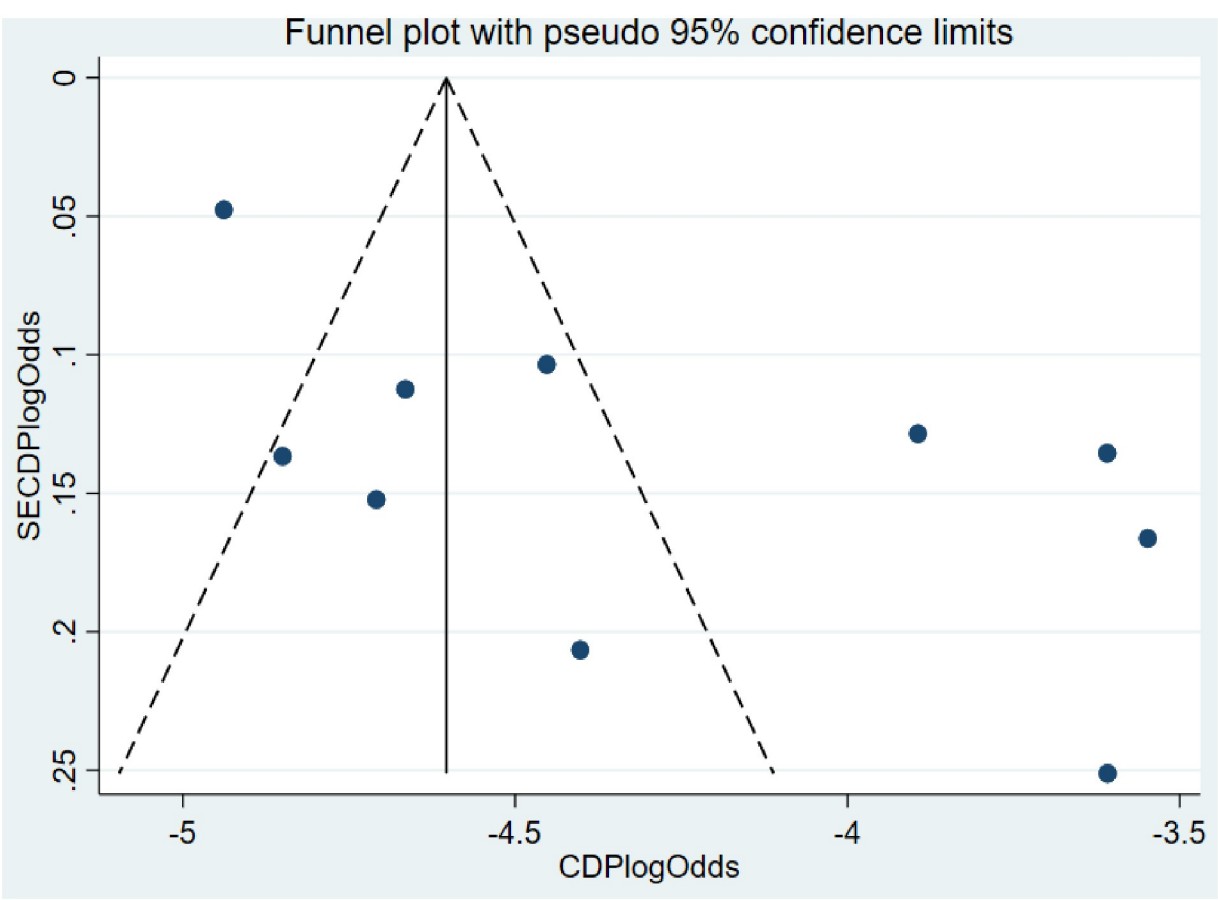

**Fig 3. Funnel plot test for publication bias for the pooled prevalence of *C. difficile* among hospitalized diarrheal patients.**

of heterogeneity as shown by the subgroup analysis ($I^2 = 0$; p = 0.96), except for studies in Iran ($I^2 = 75\%$; p = 0.007).

**Ciprofloxacin resistance of *C. difficile*.** The susceptibility to ciprofloxacin was determined in 7 studies investigating 1247 *C. difficile* isolates. From them, 973/1247(78%) isolates were found to be resistant. The WPR to ciprofloxacin was 64% (95%CI: 48–80; p<0.001) with a substantial heterogeneity ($I^2 = 98.6\%$; p< 0.001) (Table 3).

## Discussions

This systematic review and meta-analysis were conducted to estimate the pooled prevalence and antimicrobial susceptibility pattern of *C. difficile* among hospitalized diarrheal patients

**Table 2. Funnel plot (Egger's test) for publication bias among hospitalized diarrheal patients.**

| Number of studies = 10 | | | | Root MSE = 11.1 | | |
|---|---|---|---|---|---|---|
| Std_ Eff | Coef. | Std.Err. | t | P> |t| | [95% Conf. Interval] | |
| Slope | 0.0858835 | 0.131205 | 0.65 | 0.531 | -0.2166759 | 0.3884428 |
| Bias | -19.50554 | 4.501512 | -4.33 | 0.003 | -29.88604 | -9.125031 |

Test of HO = no small- study effects P = 0.003

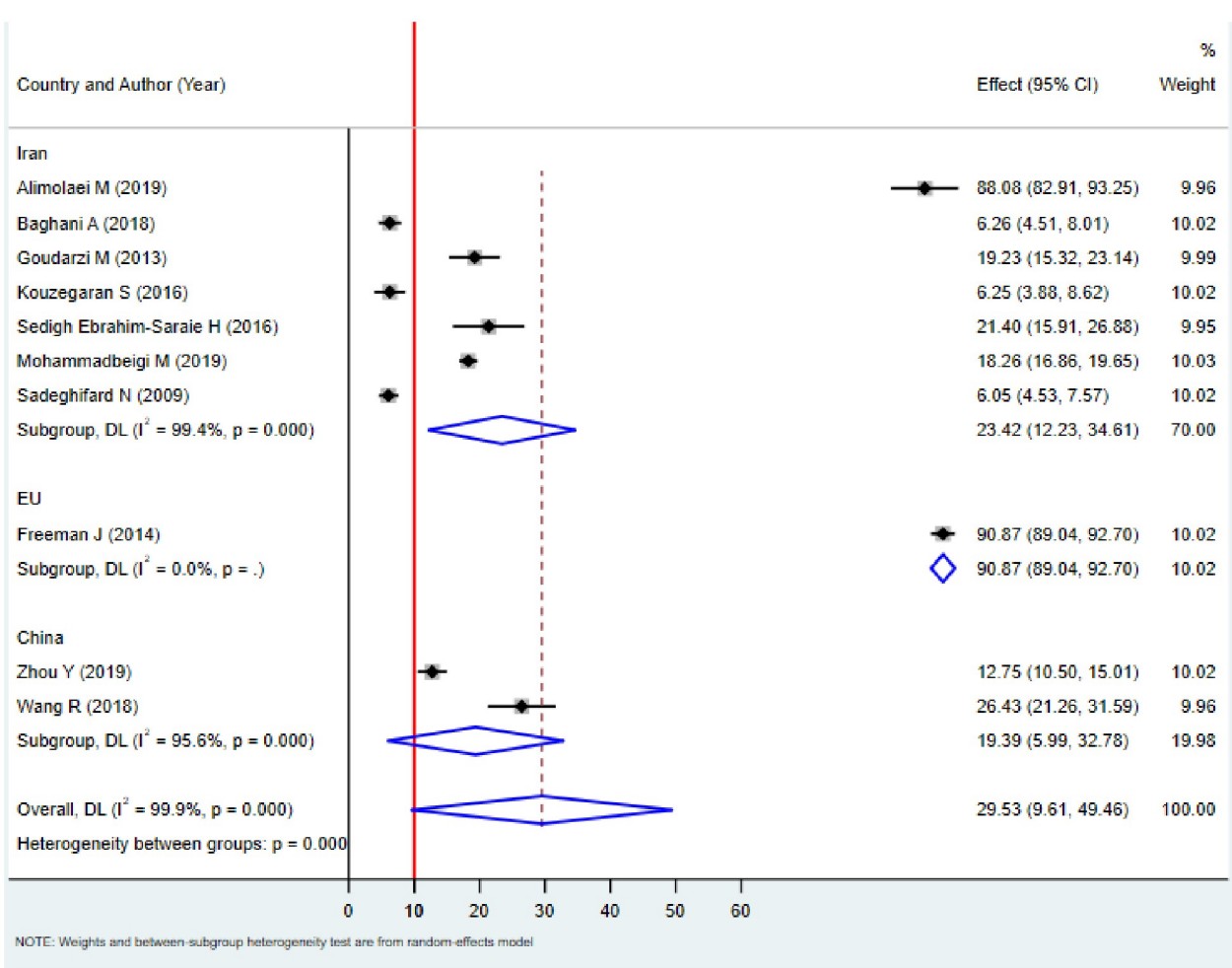

**Fig 4. Forest plots for the proportion of *C. difficile* by country among hospitalized diarrheal patients.**

worldwide. Thus, it is essential to gain a close estimation of the burden of CDI for the development of effective healthcare practice. The weighted pooled prevalence of *C. difficile* was 30.0% (Fig 2) which is higher than a study by Curcio, *et al*, [52], Asia [53], Mainland China [54], and Persian Gulf countries [55] with a pooled prevalence of 15%, 14.8%, 14%, and 9%, respectively. There was significant heterogeneity among the included studies similar to the aforementioned study in Persian Gulf countries [55]. These variations might be due to differences in predominant epidemic strains, study designs of included study, geographical distribution, studied population, or the sensitivity of detection methods. The prevalence of CDI differed greatly between studied countries.

In the current study, significant heterogeneity was observed between studies. For example, the prevalence of *C. difficile* among diarrhea patients varied in different subgroups, such as study area and year of publication. In the subgroup analysis, in Iran and China, the pooled prevalence of *C. difficile* was found to be 23.42% and 19.39%, respectively (Fig 4). Similarly, the pooled prevalence of *C. difficile* based on the year of publication increased from earliest to latest. For example, the pooled prevalence in the year 2009 was, 6.05%; in 2016, 13.61%, and in 2019, 39.52% (Fig 5). This is comparable with a previous study [56].

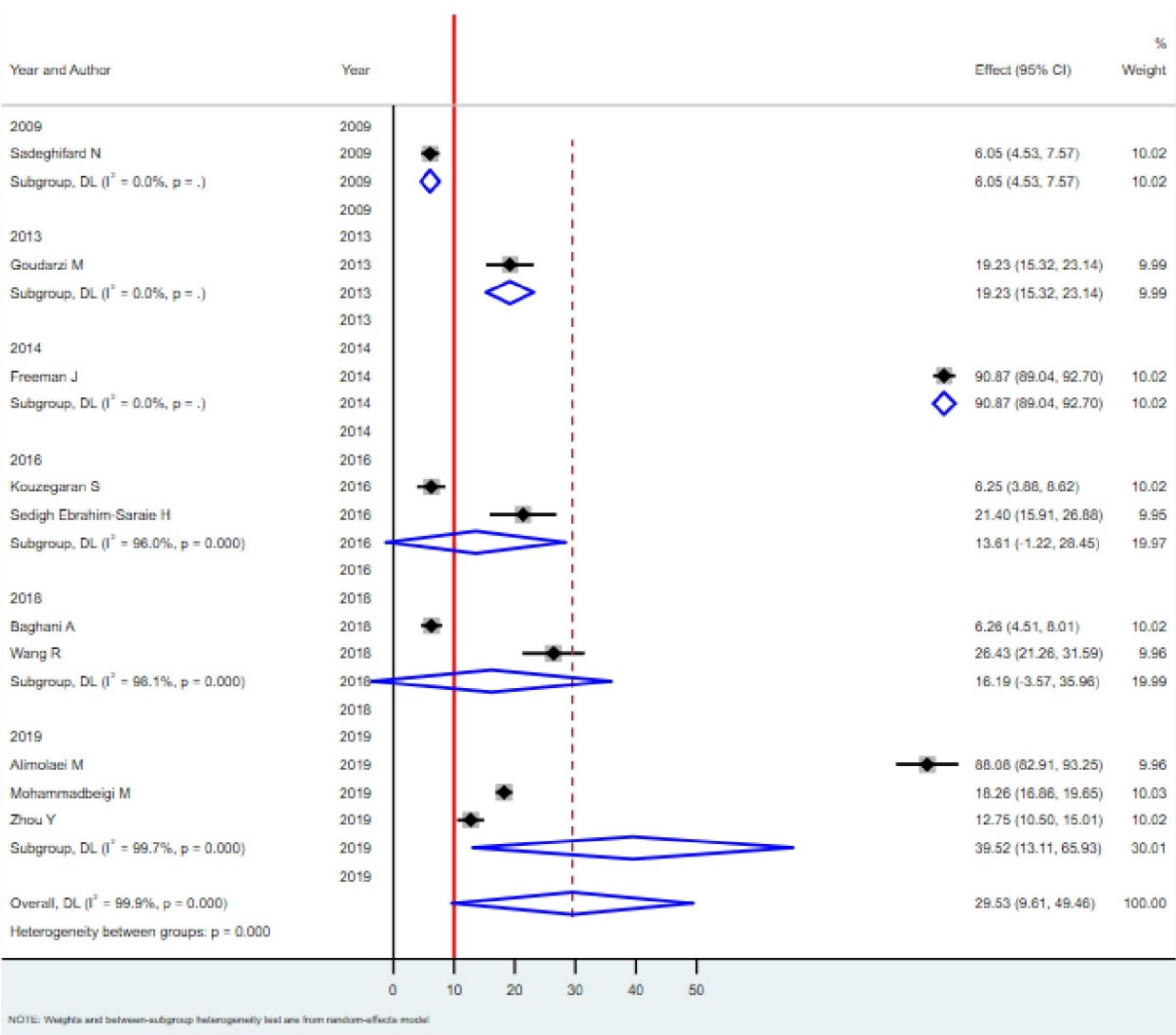

**Fig 5. Forest plots for the proportion of *C. difficile* by year of publication among hospitalized diarrheal patients.**

Metronidazole and vancomycin are two main first-line antibiotics that are used to treat primary and recurrent CDI [57]. In this review, metronidazole and vancomycin were still the drugs of choice as indicated by less pooled resistances, 5% and 3%, respectively (Table 3). The pooled resistances of *C. difficile* against metronidazole and vancomycin were 5% and 3%,

**Table 3. The Weighted Pooled Resistant (WPR) rate of *C. difficile* for each antimicrobial from hospitalized diarrheal patients.**

| Antimicrobials | Number of studies | Number of isolates | Number of resistant isolates | Weighted resistant rate (%) | 95%CI | Heterogeneity, $I^2$ |
|---|---|---|---|---|---|---|
| Vancomycin | 15 | 2755 | 114 | 3 | 1–4 | 89.5%, p<0.001 |
| Metronidazole | 15 | 2753 | 137 | 5 | 3–7 | 93.6%, p<0.001 |
| Clindamycin | 13 | 2638 | 1550 | 61 | 52–69 | 95.1%, p<0.001 |
| Moxifloxacin | 11 | 2503 | 924 | 42 | 29–54 | 97.8%, p<0.001 |
| Tetracycline | 10 | 2321 | 878 | 35 | 22–49 | 98.3%, p<0.001 |
| Erythromycin | 9 | 1544 | 987 | 61 | 48–75 | 97.3%, p<0.001 |
| Ciprofloxacin | 7 | 1247 | 973 | 64 | 48–80 | 98.6%, p<0.001 |

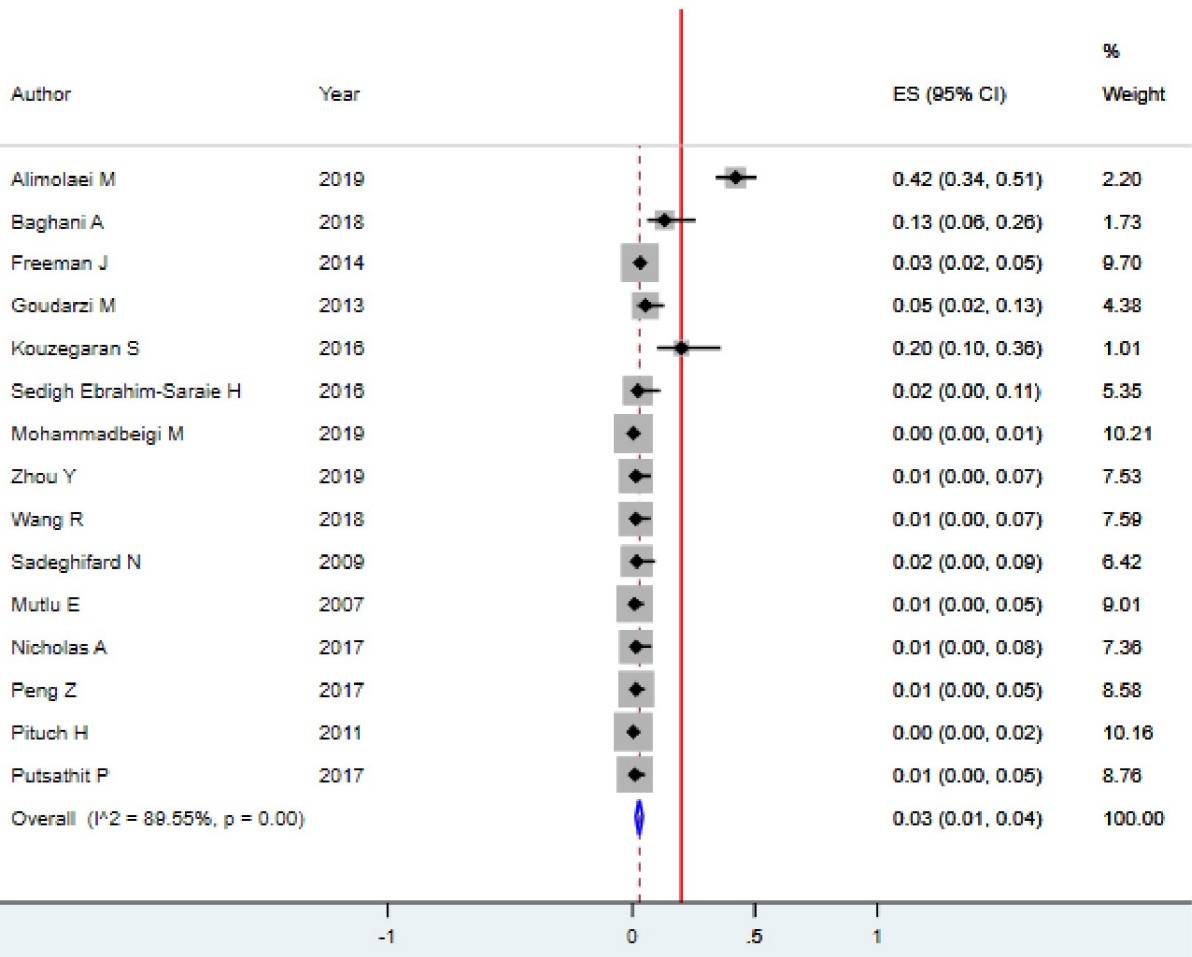

**Fig 6. Forest plot for the proportion of vancomycin resistance of *C. difficile* among hospitalized diarrheal patients.**

p<0.001), respectively. This finding is parallel to a previous study done with 1.9% for metronidazole and 2.1% for vancomycin [58], but higher than in a study in Iran with weighted pooled resistance of 1% for both drugs [56]. Another report in Iran showed that higher weighted pooled resistance of metronidazole and vancomycin were reported, 10.7% and 12.5%, respectively [59]. On contrary, no resistant strains have been identified in a study in Mainland China [54] for metronidazole and vancomycin. The difference might be variation in the year of study and publication, the origin of the isolates, the current study focused only on *C. difficile* from human origin, but others also collected data from animal origin.

In the present study, clindamycin resistance was found in 61.0% of *C. difficile* isolates which is parallel to a study by Sholeh, *et al*, 59% [56], but lower than other systematic review and meta-analysis studies in Iran and Mainland China, 84.3% and 81.7% [54, 59]. Further, erythromycin and tetracycline resistances were observed, 61% and 35% in the current study which is consistent with a study in Iran [59] with resistances, 61.5%, and 32.5%, respectively. This variation may be due to differences in the year of study, susceptibility tests used, geographic area, and strain types of *C. difficile*.

The weighted pooled resistance of moxifloxacin was 42% (95%CI: 29.0–54.0; p<0.001). A higher resistance rate was reported in Iran, 67.9% [59], but lower resistance was reported by a

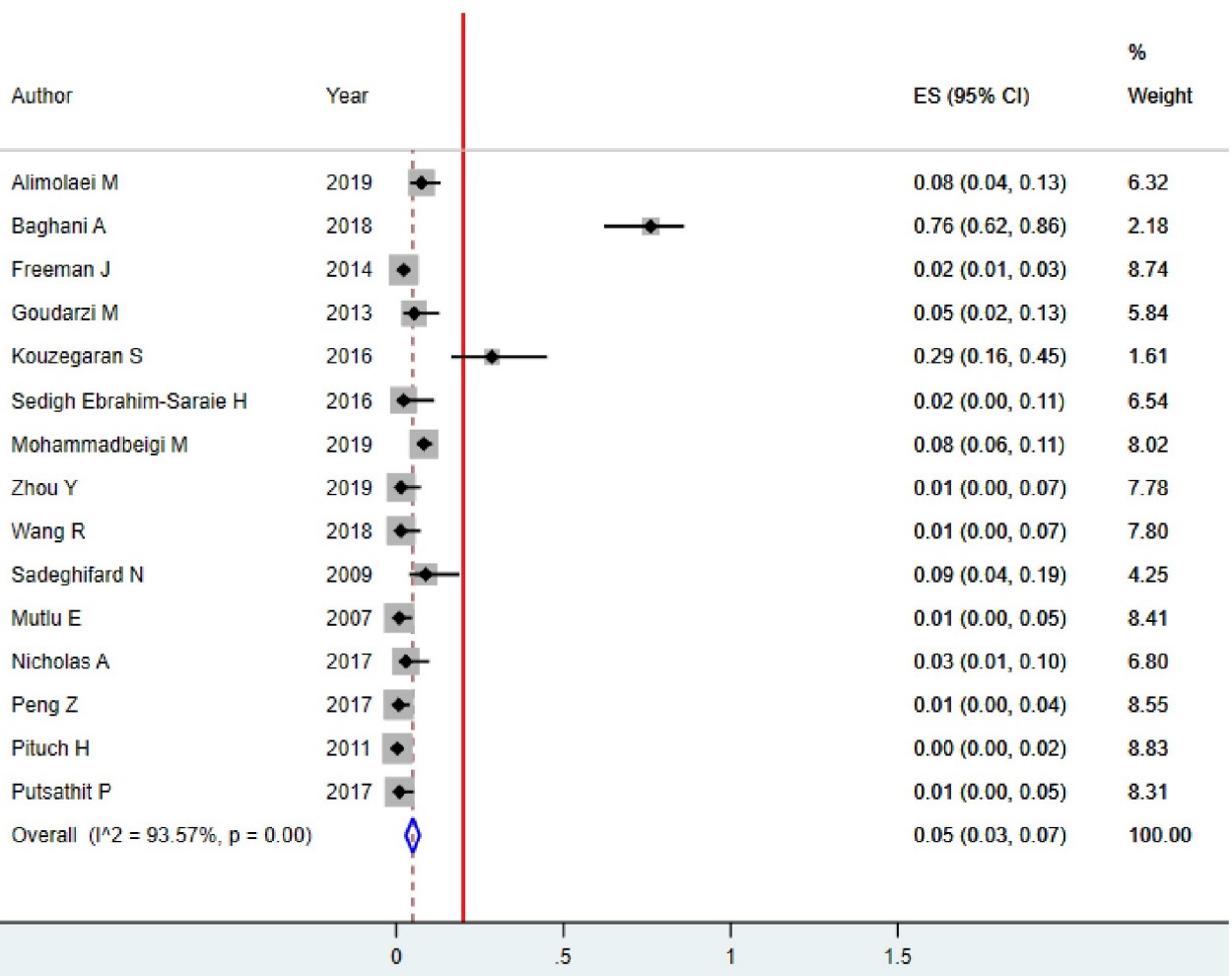

**Fig 7. Pooled proportion of metronidazole resistance of *C. difficile* among hospitalized diarrheal patients.**

systematic review and meta-analysis study by Sholeh, *et al*, 32% [56]. Similarly, the resistance of ciprofloxacin was 64%, which is in agreement with a study in Iran, 69.5% [59], but lower than a study in Mainland China, 98.3% [54]. Currently, it has been suggested that excessive use of fluoroquinolone antibiotics such as moxifloxacin and ciprofloxacin is associated with the emergence of hypervirulent *C. difficile* 027/BI/NAP1 strains [46]. The use of broad-spectrum antibiotics and the source of the isolate (human, animal, or vegetable) could contribute to variation in prevalence and antimicrobial resistance. This is supported by a published paper that there was a strong association with the use of clindamycin and cephalosporin drugs for both prevalence and resistance pattern of *C. difficile* [60].

Finally, according to the eligibility criteria of this review, most of the studies of these isolates were conducted in Iran and China (Table 1). But there is a paucity of data on the prevalence and antimicrobial susceptibility pattern of *C. difficile* from regions in Africa and South America based on this review. Even in Ethiopia, there was no single study that tells about the general description, prevalence, and antimicrobial susceptibility pattern of *C. difficile*.

## Conclusions and recommendations

The overall weighted pooled prevalence of *C. difficile* was 30% which requires action to be taken to decrease the ever-increasing infections. Vancomycin and metronidazole are still the drugs of choice to treat *C. difficile* infections (CDI) as indicated by low pooled resistance. Higher weighted resistance was observed in ciprofloxacin and clindamycin, so these drugs could not be the recommended drugs to treat CDI. Reliable information on *C. difficile* susceptibility to antibiotics could support describing emerging trends in resistance. No research was conducted which describes the burden and antimicrobial susceptibility pattern of CDI in Ethiopia. Therefore, researchers and stakeholders better give attention to this area and search more about this infection.

## Limitations

This systematic review and meta-analysis were shown *C. difficile* rate worldwide. But it may lack international representativeness because no data were found from some continents such as Africa and South America. The review included only papers that were published in English.

## Supporting information

**S1 Checklist. PRISMA 2009 checklist.**
(DOC)

**S1 Fig. Sensitivity analysis for the pooled prevalence of *C. difficile* among studies on hospitalized diarrheal patients.**
(TIF)

**S2 Fig. Subgroup analysis of vancomycin resistance by country among studies on hospitalized diarrheal patients.**
(TIF)

**S3 Fig. Subgroup analysis of vancomycin resistance by year of publication among studies on hospitalized diarrheal patients.**
(TIF)

**S4 Fig. Subgroup analysis for clindamycin resistance of *C. difficile* by country among studies on hospitalized diarrheal patients.**
(TIF)

## Author Contributions

**Conceptualization:** Tebelay Dilnessa, Workagegnehu Hailu, Feleke Moges, Baye Gelaw.

**Data curation:** Tebelay Dilnessa, Alem Getaneh.

**Formal analysis:** Tebelay Dilnessa, Alem Getaneh, Feleke Moges, Baye Gelaw.

**Investigation:** Tebelay Dilnessa, Alem Getaneh.

**Methodology:** Tebelay Dilnessa, Alem Getaneh, Workagegnehu Hailu, Feleke Moges, Baye Gelaw.

**Resources:** Tebelay Dilnessa, Alem Getaneh, Baye Gelaw.

**Software:** Tebelay Dilnessa, Alem Getaneh, Feleke Moges.

**Supervision:** Workagegnehu Hailu, Feleke Moges, Baye Gelaw.

**Validation:** Workagegnehu Hailu, Feleke Moges, Baye Gelaw.

**Visualization:** Tebelay Dilnessa.

**Writing – original draft:** Tebelay Dilnessa, Alem Getaneh.

**Writing – review & editing:** Tebelay Dilnessa, Alem Getaneh, Workagegnehu Hailu, Feleke Moges, Baye Gelaw.

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
