## [Decision Letter · Decision Letter 0]

17 Aug 2021

PONE-D-21-16139

Prevalence and antimicrobial resistance pattern of Clostridium difficile: a systematic review and meta-analysis

PLOS ONE

Dear Dr. Dilnessa,

Thank you for submitting your manuscript to PLOS ONE. After careful consideration, we feel that it has merit but does not fully meet PLOS ONE’s publication criteria as it currently stands. Therefore, we invite you to submit a revised version of the manuscript that addresses the points raised during the review process.

Two experts in your field have reviewed your manuscript. The two reviewers have suggested a major revision. Could you please follow the reviewers' comments and make the necessary revisions.  

We look forward to receiving your revised manuscript.

Kind regards,

Yung-Fu Chang

Academic Editor

PLOS ONE

Journal Requirements:

2. Thank you for submitting the above manuscript to PLOS ONE. During our internal evaluation of the manuscript, we found significant text overlap between your submission and the following previously published works, some of which you are an author.

- https://www.researchsquare.com/article/rs-4263/v1

- https://www.sciencedirect.com/science/article/abs/pii/S1046592810001282?via%3Dihub

- https://www.sciencedirect.com/science/article/abs/pii/S1075996418300234?via%3Dihub

- https://aricjournal.biomedcentral.com/articles/10.1186/s13756-020-00815-5

- https://journals.plos.org/plosone/article?id=10.1371/journal.pone.0244057

- https://www.nature.com/articles/srep37865?code=5bd7e315-eb97-43aa-965b-4f80e34626cd&error=cookies_not_supported

- https://www.tandfonline.com/doi/abs/10.1080/20477724.2019.1603003?journalCode=ypgh20

- https://www.hindawi.com/journals/ijmicro/2020/9461901/

- https://www.biomerieux.com.tr/sites/subsidiary_uk/files/c-difficile-booklet-final-update-2013_0.pdf

- https://www.sciencedirect.com/science/article/abs/pii/S0891552014000907?via%3Dihub

Please revise the manuscript to rephrase the duplicated text, cite your sources, and provide details as to how the current 

Reviewers' comments:

Reviewer's Responses to Questions

**Comments to the Author**

1. Is the manuscript technically sound, and do the data support the conclusions?

Reviewer #1: Partly

Reviewer #2: Yes

2. Has the statistical analysis been performed appropriately and rigorously? 

Reviewer #1: Yes

Reviewer #2: Yes

3. Have the authors made all data underlying the findings in their manuscript fully available?

Reviewer #1: Yes

Reviewer #2: Yes

4. Is the manuscript presented in an intelligible fashion and written in standard English?

Reviewer #1: Yes

Reviewer #2: No

5. Review Comments to the Author

Reviewer #1: This study Dilnessa et al described presents a sustematic review and meta-analysis on prevalence and antimicrobial resistance pattern of C.difficile. C.difficile is the leading cause of infectious disease that has been concerned in most of countries. The data in this study were collected from some regions in the worldwide, and the results reported here were a little bias. The comments were presented as below.

1. In line 35, C.difficile has also become a big issue in community-acquired diarrhea.

2. In lines 13-16, this paragraph should be omitted. the paragraph in lines 16-19 should be an initial paragraph in Introduction.

3. In lines 7-10, the reference 5 published in 2009, the data in ref 5. did not display the data from the last decade.

4. In lines 10-11, the authors should provide the geographical information.

In lines 105-106, this is not an exclusion criteria. The authors should involve all the published data meeting inclusion criteria.

5. As figure 1 shown, only fifteen studies was too few to meet the study. I just searched C.difficle, molecular epidemiology, antimicrobial resistance/ susceptibility pattern in the PubMed. There were more than one hundred studies meeting the inclusion criteria. I strongly recommended that the authors involved more data into this study.

Minor comments:

1. In line 6, omit the letter"a".

2. In line 33, C.difficile producing....

3. In line 44, omit the letter"s".

4. In line 118, provide the full name of JBI.

5. There are some mistakes in Table 1, e.g. Peng Z and Wang R from China.

6. Please provide high resolution pictures as the figures.

Reviewer #2: This is a systematic review and meta-analysis of the prevalence of CDI and the prevalence of resistance to common antimicrobials of CD strains among hospitalised patients with diarrhea. The topic is of health care importance.

The study background appears sometimes partly outdated as CD resistance to most commonly used antibiotics is not an emerging issue but is a reality since at least 2 decades.

The title clearly describes the main focus but doesn’t mention the specific setting, i.e. hospitalised patients with diarrhea.

The abstract sums up the main contents of the work with coherence and effectiveness. However, it mentions hypervirulent strains that are not the focus of any specific evaluation. Also, it states CDI prevalence came out ‘higher’ but it is not clear with respect to which anticipated value.

The methods section reports all information about the study, along with inclusion and exclusion criteria.

Statistical analysis is well described.

Results are not always clearly presented. In the text, it is difficult to differentiate cases (i.e. patients with the disease) of CDI from strains evaluated for susceptibility. Often the impression is that authors failed to re-read the article for a final revision. Just as an example, sentence at line 236 appears truncated (it should read ‘for estimating the pooled resistance is applicable’).

Major comments:

1. A more clinically-relevant focus of the study would be to assess CD resistance to the 3 commonly used antimicrobials (vanco, metron, fidaxom) and some newer assessed options, that are only partly mentioned by authors, but not studied. Assessing prevalence of AMR for ciprofloxacin, clindamycin or erythromycin has much less relevance as none of these molecules are used for CDI treatment.

2. Inclusion of patients with diarrhea: were these patients who were admitted because of diarrhea or developed diarrhea during hospitalization for other reasons? Were patients on any antibiotic before or at the time of diarrhea onset?

3. Beyond formal evaluation of heterogeneity of results, it is significant to note the large difference in prevalence of CDI in the EU study and in the Iran+China studies. This requires authors to describe which were the diagnostic tools used for diagnosis (culture, toxigenic culture, GDH/toxin EIA, polymerase chain reaction?). The same applies for antimicrobial susceptibility testing: which were tests used and was any heterogeneity in terms of methodology used across studies?

4. At the beginning of Results section, authors repeat that studies had features already described in Methods, inclusion/exclusion criteria. These details are not needed in the Results and make the text too long.

5. Country of studies: Scotland and Poland are in Europe. It is odd that half of the studies included came from Iran. By the way, this makes generalizability of results much more problematic, also in light of the fact that Iranian studies weight was 70%. On double checking Table 1, ref. 50 Freeman is not from Iran, ref. 45 Mohammadbeigi is not from China but Iran. … I think most data regarding countries are wrong. Referring to the weighted pooled proportion authors define it as ‘worldwide’. This is not the case as the number of countries included is limited. The notation worldwide should be essentially removed.

6. Abstract Line 34: not necessarily after hospitalization; it is usually after antibiotic administration. Hospital emerging cases are the majority as community based diagnosis is more difficult.

7. A thorough revision of the English language is needed.

6. PLOS authors have the option to publish the peer review history of their article (what does this mean?). If published, this will include your full peer review and any attached files.

Reviewer #1: **Yes: **Dazhi Jin

Reviewer #2: **Yes: **Emanuele Durante Mangoni, MD PhD

---

## [Author Response · Author response to Decision Letter 0]

6 Nov 2021

Dear Oriel Jerome Delas Alas Vida,

First, we would to acknowledge you for the comment forwardedDear,

First, we would to acknowledge you for the comment forwarded to incorporate the PRISMA checklist. Now I incorporated. But I have already submitted previously even thou it might be removed during revision. 

Finally, once again we would like to thank you.

In case of any questions and doubts, please do not hesitate to contact us anytime.

 Tebelay Dilnessa

---

## [Decision Letter · Decision Letter 1]

26 Dec 2021

PONE-D-21-16139R1Prevalence and antimicrobial resistance pattern of Clostridium difficile among hospitalized diarrheal patients: a systematic review and meta-analysisPLOS ONE

Dear Dr. Dilnessa,

Thank you for submitting your manuscript to PLOS ONE. After careful consideration, we feel that it has merit but does not fully meet PLOS ONE’s publication criteria as it currently stands. Therefore, we invite you to submit a revised version of the manuscript that addresses the points raised during the review process. Your manuscript has been reviewed by one of the previous reviewers and a minor revision  is required before a decision can be made.

We look forward to receiving your revised manuscript.

Kind regards,

Yung-Fu Chang

Academic Editor

PLOS ONE

Journal Requirements:

Reviewers' comments:

Reviewer's Responses to Questions

**Comments to the Author**

1. If the authors have adequately addressed your comments raised in a previous round of review and you feel that this manuscript is now acceptable for publication, you may indicate that here to bypass the “Comments to the Author” section, enter your conflict of interest statement in the “Confidential to Editor” section, and submit your "Accept" recommendation.

Reviewer #1: All comments have been addressed

2. Is the manuscript technically sound, and do the data support the conclusions?

Reviewer #1: Yes

3. Has the statistical analysis been performed appropriately and rigorously? 

Reviewer #1: Yes

4. Have the authors made all data underlying the findings in their manuscript fully available?

Reviewer #1: Yes

5. Is the manuscript presented in an intelligible fashion and written in standard English?

Reviewer #1: No

6. Review Comments to the Author

Reviewer #1: The revised manuscript has been improved and better than the previous one. However, the language needs to be checked and edited again. In line 370, the word "was" is corrected to "were". In line 371, The pooled resistances.....In line 376, no resistant strains have been identified...... e.g.

7. PLOS authors have the option to publish the peer review history of their article (what does this mean?). If published, this will include your full peer review and any attached files.

Reviewer #1: **Yes: **Dazhi Jin

---

## [Author Response · Author response to Decision Letter 1]

28 Dec 2021

Dear Dr. Yung-Fu Chang

Academic Editor, PLoS One

First, once again we would to acknowledge you for comments forwarded which is very important for the improvement of the manuscript. We would like to thank very much the reviewer for providing important comments for our manuscript. Then, please find the attached revised documents that were accommodated with the comments forwarded by the academic editor and reviewer. 

Responses to Academic Editor:

Academic Editor: Journal Requirements: Please review your reference list to ensure that it is complete and correct. If you have cited papers that have been retracted, please include the rationale for doing so in the manuscript text, or remove these references and replace them with relevant current references. Any changes to the reference list should be mentioned in the rebuttal letter that accompanies your revised manuscript. If you need to cite a retracted article, indicate the article’s retracted status in the References list and also include a citation and full reference for the retraction notice.

Author response: Thank you for sharing the PLOS ONE's journal requirements for references. We went through each reference and checked online on databases its availability. We got all references used in the manuscript were complete, correct and available in the databases. There are no retracted references used in this manuscript. Again, we made no any changes to the references list.

Responses to Reviewer #1:

Thank you for your comments and we accepted comments and incorporated to the manuscript.

Reviewer 1: The language needs to be checked and edited again. In line 370, the word "was" is corrected to "were". In line 371, The pooled resistances...In line 376, no resistant strains have been identified.... e.g. 

Author response: Thank you for the comment. A comprehensive English language editing, revisions and topographical corrections were made throughout the manuscript. For example:-

• Metronidazole and vancomycin were still the drugs of choice…. (line 370)

• The pooled resistances of C. difficile against…….(line 371) 

• No resistant strains have been identified…… (376)

• Further, erythromycin and tetracycline resistances were observed……. (384), 

• Etc, were revised and corrected.

Finally, once again we would like to thank academic editor and the reviewer.

 Tebelay Dilnessa

---

## [Editor Report · Decision Letter 2]

30 Dec 2021

Prevalence and antimicrobial resistance pattern of Clostridium difficile among hospitalized diarrheal patients: a systematic review and meta-analysis

PONE-D-21-16139R2

Dear Dr. Dilnessa,

We’re pleased to inform you that your manuscript has been judged scientifically suitable for publication and will be formally accepted for publication once it meets all outstanding technical requirements.

Kind regards,

Yung-Fu Chang

Academic Editor

PLOS ONE
---

## [Editor Report · Acceptance letter]

5 Jan 2022

PONE-D-21-16139R2 

Prevalence and antimicrobial resistance pattern of *Clostridium difficile* among hospitalized diarrheal patients: a systematic review and meta-analysis 

Dear Dr. Dilnessa:

I'm pleased to inform you that your manuscript has been deemed suitable for publication in PLOS ONE. Congratulations! Your manuscript is now with our production department. 

Kind regards, 

on behalf of

Dr. Yung-Fu Chang 

Academic Editor

PLOS ONE